# Text message reminders for improving sun protection habits: A systematic review

**Diego Chambergo-Michilot[1], Walter Andree Tellez[2], Naysha Becerra-Chauca[3], Jessica Hanae Zafra-Tanaka[4], Alvaro Taype-Rondan**  **[5] ***

**1** Universidad Científica del Sur, Lima, Peru, **2** Universidad Nacional Federico Villarreal, Lima, Peru,
**3** Instituto de Evaluación de Tecnologías en Salud e Investigación—IETSI, EsSalud, Lima, Perú,
**4** Universidad Peruana Cayetano Heredia, CRONICAS Center of Excellence for Chronic Diseases, Lima,
Peru, **5** Universidad San Ignacio de Loyola, Unidad de Investigación para la Generación y Síntesis de
Evidencias en Salud, Lima, Peru

* alvaro.taype.r@gmail.com

org/10.1371/journal.pone.0233220

UNITED KINGDOM

**Data Availability Statement:** All relevant data are
within the paper and its Supporting Information
files.

## Abstract

### Background

Excessive exposure to ultraviolet radiation increases the risk of skin cancer and other condi-
tions. SMS text reminders may be a useful tool to improve sun protection habits due to its
massive reach, low cost, and accessibility.

### Objective

To perform a systematic review of randomized controlled trials (RCTs) that evaluated the
effects of SMS text reminders in promoting sun protection habits.

### Methods

We performed a systematic search in PubMed, Central Cochrane Library, and Scopus; fol-
lowing the PRISMA recommendations to perform systematic reviews. We included RCTs
published up to December 2018, which evaluated the benefits and harms of SMS text
reminders to improve sun protection habits. Random-effects meta-analyses were performed
whenever possible. The certainty of the evidence was assessed for RCTs estimates using
the Grading of Recommendations Assessment, Development, and Evaluation (GRADE)
methodology. The study protocol was registered in PROSPERO (CRD42018091661).

### Results

Five RCTs were included in this review. When pooled, the studies found no effect of SMS
text reminders in "sunburn anytime during follow-up" (two studies, risk ratio: 0.93; 95% confi-
dence interval: 0.83–1.05). Contradictory results were obtained for sunscreen use (three
RCTs) and sun protection habits (two RCTs), however, they could not be meta-analyzed
because outcomes were measured differently across studies. The certainty of the evidence
was very low for these three outcomes according to GRADE methodology.

**Funding:** The authors received no specific funding for this work.

**Competing interests:** The authors have declared that no competing interests exist.

## Conclusions

RCTs that assessed effects of SMS text reminders did not find a significant benefit on objective outcomes, such as having a sunburn, sunscreen use and composite score of sun protection habits. Since certainty of the evidence was very low, future high-quality studies are needed to reach a conclusion regarding the balance of desirable and undesirable outcomes.

## Protocol registration number

PROSPERO (CRD42018091661).

## Introduction

Excessive exposure to ultraviolet (UV) radiation increases the risk for developing severe diseases, such as skin cancer (especially melanoma and basal cell carcinoma) [1], macular degeneration, and cataracts [2]. Sun protection habits to diminish this exposure include sunscreen use, wearing a hat, wearing sunglasses, wearing clothes that protects vulnerable skin areas, avoidance of exposure during peak UV hours around solar noon, and seeking shade or staying indoors [3]. Accordingly, educational interventions to enhance compliance with sun protection habits have been proposed, and considerable effects have been observed in certain groups, such as melanoma survivors, parents of young children, and medical professionals [4–6].

Evidence about the effects of mass media interventions in health outcomes, such as oral and print-based health promotion campaigns, suggests their usefulness, but is limited by the study designs and problems in the measurement of the outcome, moreover, these interventions are not usually tailored [7]. In this way, there is growing interest in technology-based interventions, such as the use of mobile applications [8], electronic mails [9] and short message service text message reminders (SMS text reminders) [10, 11]. Some systematic reviews have synthesized the evidence of the effects of SMS text reminders and mobile applications in medication adherence and management, in adolescents [12–14] and adults [15], supporting their feasibility and acceptability. Nevertheless, all of those reviews recommended future studies with a more fitting design.

Using SMS text reminders can be an appropriate strategy to improve sun protection habits due to its massive reach, low cost, accessibility in space and time, the potential for tailoring, and the possibility to interact with the sender [8, 16, 17]. Additionally, an increasing number of people are using mobile devices to obtain health information [8]. Consequently, SMS text reminders have been used in the management of several diseases, such as diabetes and asthma, and on the improvement of different habits, such as weight loss, smoking cessation, exercising or physical activity [8, 18] and sun protection [6].

The effects of SMS text reminders in sun protection habits have been evaluated through randomized controlled trials (RCTs). However, their results have not been synthesized, which hinders the decision-making process on this subject. Thus, the objective of this study was to perform a systematic review of RCTs to evaluate the effects of SMS text reminders in promoting sun protection habits.

## Methods

### Protocol and registration

We performed a systematic review and meta-analysis following the Preferred Reporting Items for Systematic Reviews and Meta-analyses (PRISMA) recommendations [19]. The study protocol has been registered at PROSPERO (CRD42018091661).

### Information sources, search and study selection

For this systematic review, we included all RCTs that directly evaluated the effects of SMS text reminders on outcomes related to sun protection habits in the SMS receivers.

Searching was performed in two steps: 1) a systematic search in three databases, and 2) a review of all documents that have cited any of the studies included in step 1, and of all the references of the studies included in step 1.

To carry out step 1, we performed a literature search in three databases: PubMed, Central Cochrane Library (CENTRAL), and Scopus. No restrictions in language or publication date were applied. The last research update was performed in December 2018. The detailed search strategy for this step is available on S1 Material. We downloaded all found references to an EndNote document, and eliminated duplicated articles using this software. After that, we assessed titles and abstracts to identify potential studies for inclusion. Lastly, we assessed the full-text of these potential studies to determine their eligibility. The complete list of articles that were excluded in the full-text assessment is detailed in S2 Material.

For step 2, during February 2019, we reviewed all documents that have cited any of the studies included in the first step, using Google Scholar (https://scholar.google.com.pe/), and we reviewed all the references of the studies included in step 1. Later, we collected all articles that met the inclusion criteria.

Both steps were performed independently by two reviewers (DCM and WAT). When disagreements occurred, they were discussed by all authors and resolved by consensus.

### Data extraction

Two independent authors (DCM and WAT) extracted the following information of the included studies into a Microsoft Excel worksheet: author, year of publication, title, population (inclusion and exclusion criteria), setting, intervention (duration, frequency, and activities), comparator (duration, frequency, and activities), time of follow-up, and effects of SMS text reminders. When disagreements were found, the full-text articles were reviewed again by the authors.

### Risk of bias and certainty of the evidence

To evaluate the risk of bias of included RCTs, we used the Revised Cochrane risk of bias tool for randomized trials version 2.0 (RoB 2) [20], published in October 2018. This tool assesses the risk of bias in five domains per outcome of interest: bias arising from the randomization process, bias due to deviations from the intended interventions (effect of assignment to intervention), bias due to missing outcome data, bias in measurement of the outcome, and bias in selection of the reported results; and one overall judgment. For each of the domains, the overall risk of bias (low risk, some concerns, and high risk) was established according to the judgment of their *signaling* questions.

To assess the certainty of the evidence for each outcome, we used The Grading of Recommendations Assessment, Development and Evaluation (GRADE) methodology [21], which evaluates the risk of bias, inconsistency, indirectness, imprecision and publication bias.

## Statistical analysis

For each outcome of each study, we calculated and reported the intervention effects as mean differences (MDs) or risk ratios (RRs) along with their 95% confidence intervals (95% CIs).

When two or more studies presented the same outcome in a similar fashion, we performed a meta-analysis using random-effects models (Mantel-Haenszel method) due to heterogeneity across studies interventions [22]. Meta-analyses were performed using Review Manager Software Version 5.3.

We assessed heterogeneity using the $I^2$ statistics, and we considered that heterogeneity might not be important when $I^2 < 40\%$ [23]. Publication bias was not assessed since the number of studies pooled for each meta-analysis was less than ten [23].

# Results

## Studies selection

We found 1,333 records in databases searching. After duplicates removal, we screened 1,092 records, from which we reviewed 34 full-text documents, and finally included six papers [24–29]. Later, we searched documents that cited any of the initially included studies as well as the references of the initially included studies. However, no extra articles that fulfilled inclusion criteria were found in these searches (Fig 1).

The six included papers reported results of five RCTs since two papers reported results from the same RCT: Youl 2015 [26] and Janda 2013 [27]. We will cite only Youl's paper to refer to this study since it was the one that presented results of interest.

## Characteristics

Of these five RCTs, three studies [24, 25, 29] were performed in the United States and two studies [27, 28] in Australia. Two studies [24, 28] were performed during summer months according to its hemisphere. Regarding the population characteristics, three studies [27–29] were performed in community dwellers; the female proportion ranged from 8.4% to 100%, and the mean age ranged from 31.6 to 44.2 years.

Regarding the intervention, it consisted of delivering SMS text reminders on sun protection habits, such as sunscreen use and skin self-examination, with heterogeneous frequency (range: three to seven days) and duration (range: one to twelve months). Of the five RCTs, two of them tailored the messages according to the baseline characteristics of participants, using the health belief model constructs [24, 25].

The control group received a 30-minutes educative PowerPoint presentation in one study [24], while in the other four studies [25, 26, 28, 29] this group received SMS text reminders about other topics, such as physical activity or sex protection (Table 1 and S3 Material).

## Risk of bias

The risk of bias was assessed using the RoB 2. Regarding the randomization process, three studies had some concerns, while three studies had a high risk of bias in the measurement of the outcome. Only one study had a low risk of bias in most domains [29]. Four studies had a high risk of bias in the overall judgment (Fig 2).

## Outcome effects

Several objective outcomes were reported by the included studies. Some of them were the number of sunburns, sunscreen use, sun protection habits (protecting clothes, wearing

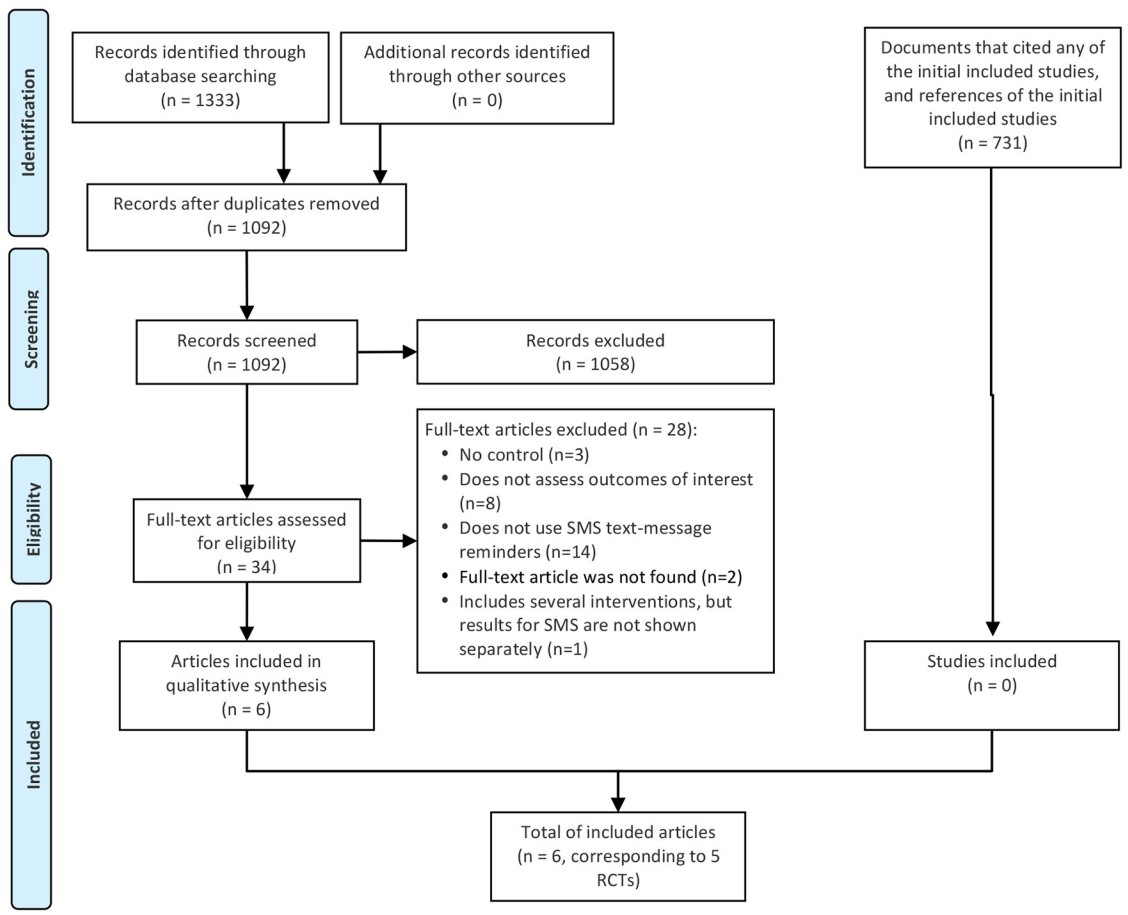

**Fig 1. Flowchart of study selection.**

sunglasses, wearing a hat, etc.), skin self-examination, attempted suntan and adherence rate of sunscreen application.

In Darlow's study [25], only two results are briefly presented in the paper text, but the number of participants in each group could not be extracted, so we could not calculate MDs' confidence intervals (Table 2).

Although many outcomes were assessed, only *sunburn anytime during follow-u*p was measured similarly in two studies [24, 26], so we could perform a meta-analysis. Although Duffy´s study had two comparisons, we used only the one that did not include mailed sunscreen for meta-analysis, because we considered that this comparison was the most similar to the result of the other study. The meta-analysis showed a pooled RR of 0.93 (95% CI: 0.83 to 1.05) (Fig 3). This result had a very low certainty of evidence (Table 3).

Sunscreen use was assessed in three studies [24, 28, 29], but was measured in different forms, so meta-analysis could not be performed. Of these studies, only Armstrong found a statistically significant benefit [29]. This study had a shorter follow-up (six weeks versus 4–5 months in the other two studies), a smaller population (70 versus 358–535 in the other two studies), but it used a more objective measurement of the sunscreen use: sending a message to the study central when the cap of the container of sunscreen was removed by the participant, while the other studies just assessed the self-report of sunscreen use. However, this result had a very low certainty of evidence (Table 3).

**Table 1. Study and participants' characteristics in the included RCTs.**

| First author, year (country) | Design | Study settings | Follow-up period | Intervention details | Funding |
|---|---|---|---|---|---|
| Duffy, 2018 (USA) [24] | Parallel RCT | Outdoor workers | 5 months | Messages guided by the health belief model constructs | The Blue Cross Blue Shield of Michigan Foundation |
| Darlow, 2017 (USA) [25] | Parallel RCT | Young adult women from a metropolitan region of the USA northeast | 4 weeks | Messages guided by the health belief model constructs | The Aetna Foundation |
| Youl, 2015 & Janda, 2013 (Australia) [26, 27] | Parallel RCT | Community dwellers (participants from the Queensland electoral and Medicare rolls) | 3 & 12 months | Personalized messages based on the social cognitive theory, which used a conversational tone | Australian National Health and Medical Research Council (NHMRC) |
| Gold, 2011 (Australia) [28] | Parallel RCT | Community dwellers | 4 months | Humorous, short, used informal language and were linked to particular annual events where possible | • VicHealth Discovery Grant<br>• The Australian Government<br>• Monash University Faculty of Medicine<br>• NHMRC |
| Armstrong, 2009 (USA) [29] | Parallel RCT | Community dwellers | 6 weeks | Two components: a text detailing daily local weather information and a text reminding users to apply sunscreen | Information Systems Council of Massachusetts General Hospital and Brigham and Women's Hospital |

*First arm**: SMS + mailed sunscreen + education, versus mailed sunscreen + education. *Second arm**: SMS + education, versus education only

SSE: skin self-examination

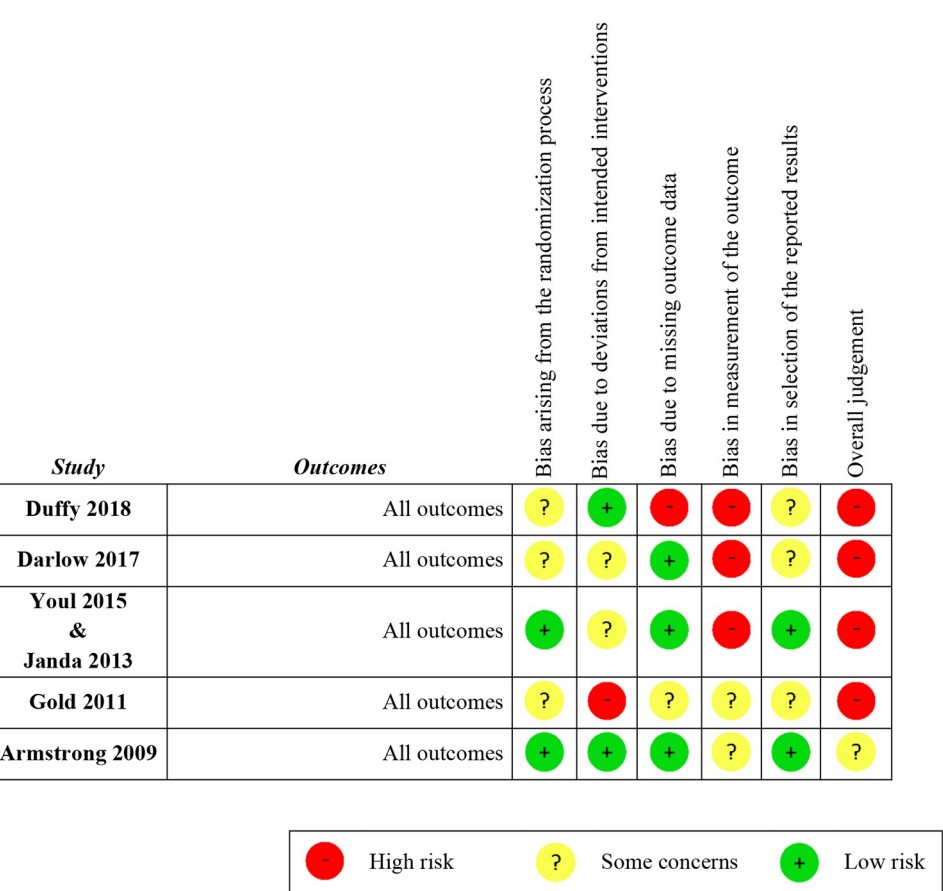

**Fig 2. Risk of bias of the included studies.**

**Table 2. Characteristics of the included studies.**

| First author, year (country) | Number of participants at baseline (intervention/ control) | Frequency and duration of SMS | Gender (female %) | Mean age in years | Follow- up | Outcomes effects |
|---|---|---|---|---|---|---|
| Duffy, 2018 (USA) [24] | First comparison: 93/87, second comparison: 86/91[1] | Three random weekdays during 5 months | 8.4% | 44.2 | 5 months | • At least one sunburn:<br>  • First comparison[1]: RR: 1.09 (0.84–1.41)<br>  • Second comparison[1]: RR: 0.85 (0.68–1.08)<br>• Sunscreen use (5-point Likert scale: from never to always):<br>  • First comparison[1]: MD: 0.1 (-0.3 to 0.5)<br>  • Second comparison[1]: MD: 0.2 (-0.2 to 0.6)<br>• Number of sunburns:<br>  • First comparison[1]: MD: 0.0 (-0.4 to 0.4)<br>  • Second comparison[1]: MD: -0.2 (-0.6 to 0.2) |
| Darlow, 2017 (USA) [25] | 104 participants distributed in 4 groups | Every day during 2 weeks | 100% | Not reported | 4 weeks | The paper does not bring enough information to assess the effects of SMS in any outcome |
| Youl, 2015 & Janda, 2013 (Australia) [26, 27] | 187/183 | Weekly for the first 3 months and monthly during the following 9 months | 67% | • 31.6 (sun protection),<br>• 31.8 (control) | 3 months | • Sun protection habits index (4-points Likert scale): MD: 0.02 (-0.07 to 0.11)<br>• Any skin self-examination (SSE) in the past 3 months: RR: 1.13 (0.84 to 1.51)<br>• Whole-body SSE at time of the last SSE: RR: 1.02 (0.66 to 1.57) |
| | | | | | 12 months | • Sun protection habits index (4-points likert scale): **MD: 0.13 (0.03 to 0.23)**<br>• Any skin self-examination (SSE) in past 3 months: RR: 1.22 (0.95–1.56)<br>• Whole-body SSE at time of last SSE: RR: 1.27 (0.72 to 2.25)<br>• Any sunburn in the past 12 months: RR: 0.96 (0.84 to 1.10)<br>• Two or more sunburns in past 12 months: RR: 0.88 (0.67 to 1.16)<br>• Attempted suntan in past 12 months: RR: 0.95 (0.58 to 1.57) |
| Gold, 2011 (Australia) [28] | 200/158 | Fortnightly during 4 months | 39.9% | Not reported | 4 months | • Preference for a dark tan: RR: 1.13 (0.59 to 2.16)<br>• Consideration of the long-term consequences of prolonged UV exposure: RR: 1.01 (0.84 to 1.20)<br>• Usually/always wears hat: RR: 1.11 (0.80 to 1.52)<br>• Usually/always wears sunscreen: RR: 0.95 (0.73 to 1.23)<br>• Usually/always seeks shade: RR: 1.00 (0.77 to 1.31)<br>• Usually/always wears deliberately skimpy clothing: RR: 0.96 (0.66 to 1.41) |
| Armstrong, 2009 (USA) [29] | 35/35 | Every day during 6 weeks | 70% | 32.9 (SMS) / 34.3 (control) | 6 weeks | Adherence rate of sunscreen application: **MD: 11 (6.5 to 15.5)** |

[1]***First*** **comparison:** (intervention: SMS + mailed sunscreen + education) vs (control: mailed sunscreen + education). ***Second*** **comparison:** (intervention: SMS + education) vs (control: education only).

The risk ratios (RRs) and mean differences (MDs) were unadjusted.

Statistically significant results were in bold.

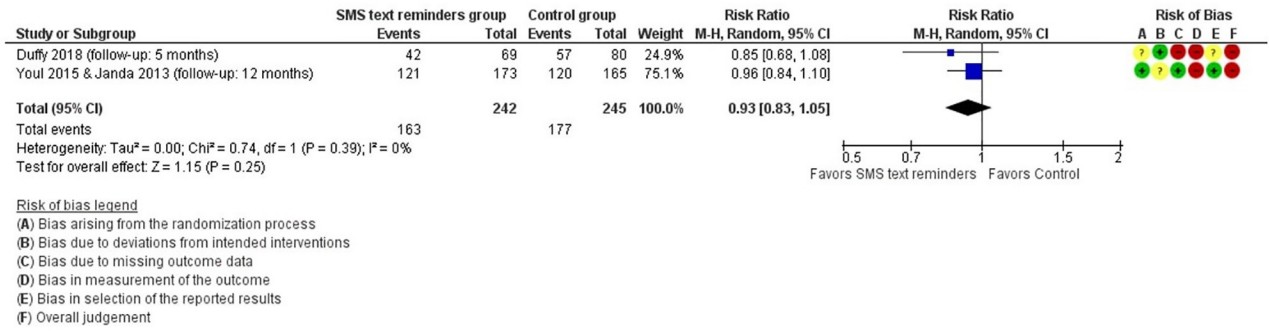

**Fig 3. Forest plot for the effect of SMS messages in having a sunburn anytime during follow-up.**

Composite sun protection habits were assessed in two studies [26, 28]. One of them used sun protection habits index claims (score range 1–4) and found no difference at three months of follow-up, but found a slight difference at 12 months (2.63 vs 2.50 on average). The other study did not find differences in the assessed outcomes (wearing a hat, seeking shade, or using skimpy clothing). This result had a very low certainty of evidence (Table 3).

Additionally, we found other outcomes that were not included in the Summary of Findings (Table 3): having a sunburn, number of sunburns, UV radiation exposure behaviors, wearing a hat, skin self-examination, attempted suntan, believe about risk of cancer, usually/always seeking shade, usually/always wearing skimpy clothes and adherence of sunscreen application. Only the UV radiation exposure behaviors and adherence to sunscreen application showed little differences between groups. The outcome effects are presented in Table 2.

## Discussion

### Summary of the results

We found five RCTs, all of them were performed in the United States or Australia. Studies had high variability in terms of interventions, control groups, assessed outcomes, and follow-up period. In the overall assessment, four studies had a high risk of bias and one study had some concerns.

We only could meta-analyze one outcome (having a sunburn anytime during the follow-up), in which we did not find a statistically significant effect. Among the three studies that assessed sunscreen use, only one had a statistically significant effect (the one with the lowest population). Among the two studies that assessed a composite score of sun protection habits, one did not find an effect while the others find a small statistically significant difference. The certainty of the evidence was very low for these outcomes. Altogether, these studies do not show compelling evidence of any beneficial effect of text message reminders.

### Previous systematic reviews

Although we did not find previous systematic reviews that have assessed the effects of SMS for sun protection habits, we found two previous systematic reviews that have assessed similar questions.

The first systematic review [11] assessed the effectiveness of SMS text reminders and similar electronic technology interventions to promote skin cancer prevention. This review describes the results of its included studies without performing any meta-analysis and concluded that there was a lack of effect of their assessed interventions in skin cancer prevention outcomes.

**Table 3. Summary of findings.**

| Outcomes | Absolute anticipated effects (95% CI) | | Relative effect (95% CI) | No. of participants (Trials) | Certainty of the evidence (GRADE) |
|---|---|---|---|---|---|
| | Control group | Intervention group | | | |
| **Sunburn anytime during follow-up** (follow-up: from 5 to 12 months) | 722 per 1,000 | 672 per 1,000 (600 to 759) | RR 0.93 (0.83 to 1.05) | 487 (2 RCTs: Duffy, Youl) (24, 26) | ⊕○○○ VERY LOW [a,b] |
| **Sunscreen use** (follow-up: 6 weeks to 5 months) | Of the three RCTs that assessed this outcome, only one (Armstrong, the one with the lowest population) found a significant difference between intervention and control group. | | | 785 (3 RCTs: Duffy, Gold, & Armstrong) (24, 28, 29) | ⊕○○○ VERY LOW [c,d] |
| **Sun protection habits** (follow-up: 3 months to 12 months) | Two studies:<br>• The first study used a composite score of sun protection habits index (score range: 1–4), found no difference at 3 months of follow-up, but found a slight difference at 12 months (2.63 vs 2.50 on average)<br>• The second study did not find differences in wearing a hat, seeking shade, or using skimpy clothing | | | 728 (2 RCTs: Youl & Gold) (26, 28) | ⊕○○○ VERY LOW [a,d] |

CI: Confidence Interval; MD: Mean difference; RR: Risk ratio; RCT: randomized controlled trial

For the sunscreen use outcome, a narrative summary of the evidence was performed, since each study assessed the outcome differently

Explanations:

[a]. We rated down two levels for risk of bias since the two RCTs had high risk of bias in the overall judgment

[b]. We rated down one level for imprecision due to the small number of participants that presented the outcome (less than 400)

[c]. We rated down two levels for methodological limitations, since two of the three RCTs had a high risk of bias in the overall judgment, and the other one had some concerns

[d]. We rated down one level for indirectness, since the RCTs differ in terms of duration of intervention and in how outcomes were measured.

This conclusion is similar to ours, although our systematic review only included studies that have assessed SMS, and our outcome definition was broader.

The second systematic review [12] explored the effectiveness of SMS text reminders and mobile applications to improve adherence to preventive behaviors (sun protection, mental attention, the continuation of contraceptive pills, the use of condoms, among others) in adolescents. It included experimental and pre-post studies and did not formulate a clear conclusion regarding SMS for sun protection habits.

## Characteristics of SMS text reminders

Personalization of the SMS content is an important element to ensure the engagement with the intervention [30], since it may influence the attitudes, motivation, and attention of the participants [31]. Among our included studies, two studies (Darlow and Gold) [25, 28] developed the intervention messages using focus group feedback, while the Youl study [26] collected a pilot survey to estimate the participants' preferences of the SMS text messages content.

For tailoring, two studies (Darlow and Duffy) [24, 25] used the health belief model profile of their participants to create the messages. The health belief model profile consists of four constructs: perceived susceptibility to ill-health, perceived severity of ill-health, perceived benefits of behavior change, and perceived barriers to taking action [32]. Since these psychological aspects can influence the participant's perception of the message, tailored messages using the health belief model profile are thought to have a greater impact.

In the included studies, the frequency of SMS ranged from every day to weekly. Although there are no uniform recommendations regarding this topic, some authors have proposed that

high-frequency interventions (at least five SMS per week) or tailored frequency could be suit-able [33, 34].

However, since the studied intervention and the outcome assessment in our systematic review were very heterogeneous, we could not assess the efficacy of health-belief-model tai-lored SMS or intervention frequency/duration. Future research is needed in order to evaluate and compare these hypotheses.

## Implications for practice

In order to describe the rationale for going from evidence to recommendation, we have assessed the criteria suggested by GRADE: balance of desirable and undesirable outcomes, fea-sibility, resource use, and certainty of evidence [35].

Regarding the balance of desirable and undesirable outcomes, we found no benefit for sun-burn and contradictory results for sunscreen use and sun protection habits. In addition, SMS text reminders may have some potential harms that were not assessed in the included studies, such as excessive discomfort, fear, anxiety, or great decrease of sun exposure in susceptible par-ticipants which could alter their metabolism (which, although rare, is possible) [36]. For exam-ple, a study in 3,194 Danish found that seeking shade and wearing protective clothing was significantly associated with lower vitamin D levels in adults [37], and an analysis of a repre-sentative survey in USA (US National Health and Nutrition Examination Survey 2003–2006) found that staying in the shade and wearing long sleeves were significantly associated with lower 25(OH)D levels, especially in individuals who reported frequent use of shade on sunny days [38].

Regarding feasibility, a high percentage of people from developed countries have a mobile phone: 86.2% of Canadian inhabitants in 2015 [39], 95% inhabitants of the United States in 2013 [40], and 94% of Australian inhabitants in 2018 [41].

Regarding resource use, systematic reviews of economic evaluations of text messaging interventions found no comprehensive evidence [42, 43]. Particularly, cost studies of SMS reminders for improving sun protection habits are needed [33], such as those performed in SMS interventions for other topics such as diabetes mellitus prevention [44], improvement of antiretroviral therapy adherence [45], and smoking cessation [46].

Regarding the certainty of the evidence, the very low certainty of our results suggests that well-designed RCTs are needed in order to provide reliable estimates. However, researchers must reflect on the need for performing more studies using SMS, since it could be descending in favor of other messengers like WhasApp. On the other hand, nowadays the ubiquitous use of smartphones may allow to use other tools such as ad-hoc mobile applications (which allows face-to-face interactions, a more friendly interaction, and using videos/images) or messengers like WhatsApp (which allows including pictures, videos, audio information) [47, 48]. Given these alternatives, maybe studies using SMS should be limited to those contexts where smart-phones use is still low.

Altogether (no clear benefits and unmeasured potential harms, lack of cost data, and very low certainty of the evidence; although high feasibility), SMS interventions use could not be recommended for improving sun protection habits.

## Limitations

Since the search was only performed in three databases, we might not have found all published studies. However, we manually searched potential studies for inclusion in the references of the included studies and searched for studies that cited our included studies in Google Scholar; which could ensure that all relevant studies are included, even those from grey literature.

Moreover, it has been evidenced that adding more databases to PubMed just increases the number of trials in 2.4% [49], and in most cases, it does not change the conclusion of the review [50].

Additionally, the body of evidence presents important limitations: 1) the studies are heterogeneous in several aspects, such as the follow-up period (varies between one month and 12 months), which difficult the comparability of their results. In fact, the performed meta-analysis pooled the result of a 5-months follow-up with the result of a 12-months follow-up. 2) Most of the outcomes were not measured in the same way. 3) Most studies measured outcomes as self-report, introducing a recall bias. 4) The studies had a high risk of bias or some concerns in several domains. 5) Overall, the certainty of the evidence was low in the main outcomes, mainly due to the risk of bias, inconsistency, and small sample size.

## Conclusion

In conclusion, we found five RCTs with high variability in terms of interventions, control groups, assessed outcomes, and follow-up time. The meta-analysis performed showed no difference in sunburn anytime during follow-up and contradictory results were seen for sunscreen use and sun protection habits (very low certainty of the evidence). High-quality studies and cost information are needed to conclude regarding the balance of desirable and undesirable outcomes.

## Supporting information

**S1 Checklist. PRISMA 2009 checklist.**
(DOC)

**S1 Material. Search strategy.**
(DOCX)

**S2 Material. Studies that were evaluated in full-text, and were excluded.**
(DOCX)

**S3 Material. Study characteristics of individual studies, in detail.**
(DOCX)

**S1 Table. Search strategy.**
(DOCX)

**S2 Table. Studies that were evaluated in full-text, and were excluded.**
(DOCX)

**S3 Table. Study characteristics of individual studies, in detail.**
(DOCX)

**S1 Database.**
(DOCX)

## Acknowledgments

DCM thanks Maria C. and Fiorella A. for their support.

## Author Contributions

**Conceptualization:** Alvaro Taype-Rondan.

**Data curation:** Diego Chambergo-Michilot, Walter Andree Tellez.

**Formal analysis:** Diego Chambergo-Michilot, Walter Andree Tellez, Jessica Hanae Zafra-Tanaka.

**Funding acquisition:** Diego Chambergo-Michilot, Walter Andree Tellez, Naysha Becerra-Chauca, Jessica Hanae Zafra-Tanaka, Alvaro Taype-Rondan.

**Investigation:** Diego Chambergo-Michilot, Walter Andree Tellez, Naysha Becerra-Chauca, Jessica Hanae Zafra-Tanaka, Alvaro Taype-Rondan.

**Methodology:** Diego Chambergo-Michilot, Walter Andree Tellez, Jessica Hanae Zafra-Tanaka, Alvaro Taype-Rondan.

**Project administration:** Diego Chambergo-Michilot, Walter Andree Tellez, Naysha Becerra-Chauca, Jessica Hanae Zafra-Tanaka, Alvaro Taype-Rondan.

**Resources:** Diego Chambergo-Michilot, Walter Andree Tellez, Naysha Becerra-Chauca, Jessica Hanae Zafra-Tanaka, Alvaro Taype-Rondan.

**Software:** Diego Chambergo-Michilot, Walter Andree Tellez, Naysha Becerra-Chauca, Jessica Hanae Zafra-Tanaka, Alvaro Taype-Rondan.

**Supervision:** Diego Chambergo-Michilot, Walter Andree Tellez, Naysha Becerra-Chauca, Jessica Hanae Zafra-Tanaka, Alvaro Taype-Rondan.

**Validation:** Diego Chambergo-Michilot, Walter Andree Tellez, Naysha Becerra-Chauca, Jessica Hanae Zafra-Tanaka, Alvaro Taype-Rondan.

**Visualization:** Diego Chambergo-Michilot, Walter Andree Tellez, Naysha Becerra-Chauca, Jessica Hanae Zafra-Tanaka, Alvaro Taype-Rondan.

**Writing – original draft:** Diego Chambergo-Michilot.

**Writing – review & editing:** Diego Chambergo-Michilot, Walter Andree Tellez, Naysha Becerra-Chauca, Jessica Hanae Zafra-Tanaka, Alvaro Taype-Rondan.

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
