## [Decision Letter · Decision Letter 0]

3 Dec 2019

PONE-D-19-24279

Text message reminders for improving sun protection habits: a systematic review

PLOS ONE

Dear Dr Taype-Rondan,

Thank you for submitting your manuscript to PLOS ONE. After careful consideration, we feel that it has merit but does not fully meet PLOS ONE’s publication criteria as it currently stands. Therefore, we invite you to submit a revised version of the manuscript that addresses the points raised during the review process.

We would appreciate receiving your revised manuscript by 17th January 2020. To enhance the reproducibility of your results, we recommend that if applicable you deposit your laboratory protocols in protocols.io, where a protocol can be assigned its own identifier (DOI) such that it can be cited independently in the future. For instructions see: http://journals.plos.org/plosone/s/submission-guidelines#loc-laboratory-protocols

We look forward to receiving your revised manuscript.

Kind regards,

Jennifer A Hirst, DPhil

Academic Editor

PLOS ONE

Journal Requirements:

Additional Editor Comments (if provided):

Abstract – why could results not be meta-analysed?

Need to update PROSPERO page

Please remove HBM acronym

Methods

Statistical analysis – this section needs correcting – lines 114-115 – “We present our results using mean differences (MD), risk ratios (RR), odds ratios (OR), and, when we meta-analyzed results from different studies for the same outcome that were measured using different scales, standardized mean differences (SMD).”

Meta-analysis was not SMD, but combined RR, MD and RR are not presented in the table only percentages for intervention and control groups or odds ratios – please be consistent and calculate OR for each outcome for each study to allow the reader to make comparisons. Be clear whether OR were adjusted or unadjusted.

Lines 117-118: There was only one outcome which was meta-analysed. Were these methods on use of repeated measures used? It looks like it wasn’t as the outcome in Table 3 is “Sunburn anytime during follow-up (follow-up: from 5 to 12 months)”. Please ensure that the methods reflect exactly what you did in the analysis and how it was reported.

Results

Was Darlow’s study actually excluded? If results were not considered to be reliable and were not used then it should be excluded and should not appear in Tables 2 & 3, the flow chart or anywhere else in the text or abstract. If you do choose to include it, then please be clear about the reasons for this.

There is a lot of information in Table 2 – it would be helpful if the main points were summarised in the text.

p-values should be included in Table 2

Overall did text messages improve outcomes or not.

Consider a table with positive outcomes in green, negative in red and non-significant left white

Or a forest plot showing direction of effect and significance for each outcome without any pooling of data to give a visual picture of whether the intervention may be beneficial

Discussion

This needs more structure. Please provide quantitative results in the summary paragraph to report effect size and p-value. Please also clearly state the other outcomes reviewed and clarify that there was no compelling evidence to suggest that text messaging resulted in any improvements.

In the “Previous systematic reviews” section, please compare the outcomes assessed and findings of these reviews with those in the current study. What does this review offer over and above the previous reviews.

A limitations section is needed and needs to include the low quality of evidence, small numbers of studies and heterogeneity in outcome reporting.

Reviewers' comments:

Reviewer's Responses to Questions

**Comments to the Author**

1. Is the manuscript technically sound, and do the data support the conclusions?

Reviewer #1: Yes

Reviewer #2: Yes

2. Has the statistical analysis been performed appropriately and rigorously? 

Reviewer #1: Yes

Reviewer #2: Yes

3. Have the authors made all data underlying the findings in their manuscript fully available?

Reviewer #1: Yes

Reviewer #2: Yes

4. Is the manuscript presented in an intelligible fashion and written in standard English?

Reviewer #1: Yes

Reviewer #2: Yes

5. Review Comments to the Author

Reviewer #1: This is an interesting systematic review that examined the evidence for text messaging as an intervention to promote adherence to sun protection strategies. Below are concerns/comments for the authors to consider.

- The authors should clarify in the abstract and the methods section if they followed reporting guidelines, such as PRISMA, or other guidelines. Include appropriate citations as well.

- Introduction section, to support the rationale for the review, the authors should include additional recent promising evidence that support feasibility, acceptability and efficacy of digital interventions for behavior change (References - PMID: 30026178; PMID: 25803705; PMID: 28506955; PMID: 26831740; PMID: 28428157; PMID: 26701961; PMID: 29273573).

- Results, the authors should include much more details for the included studies, such as age, gender, study settings, follow up, outcomes, intervention details, and other RCT related factors (blinding, randomization, etc.). Similar details should be also added to Table 1.

- All the figures are fuzzy and unclear. Replace with more clear ones.

- Discussion, the authors should expand and elaborate more on how their findings support or contrast available literature and provide suggestions for future research directions that would address existing knowledge gaps.

- Discussion, it is critical to discuss the value of including direct patients' input in the development of mhealth interventions and other key considerations for end users should be sought early on in the process of app or digital behavioral intervention design to ensure long and short term engagement (PMID: 29273573; PMID: 26844685; PMID: 27966189; PMID: 28241759).

- Discussion, the authors should also acknowledge the lack of economic data to support the use of mhealth behavioral interventions to date (PMID: 27780795; PMID: 28152012).

Reviewer #2: The manuscript describes conduct and results of a systematic review of RCTs adressing the effect of SMS text reminders in promoting sun protection measures. Overall, the manuscript represents sound work.

Specific comments:

- l. 50-52: Avoidance of exposure during peak UV hours around solar noon is mentioned by all guidelines promoting sun protection and should be included in the list of sun protection habits.

- l. 53: There are a lot more studies evaluating compliance with recommendations for sun protection. You should add recent papers (e.g. Vogel et al. Cancer Epidemiol Biomarkers Prev 2017, Barkin et al. JAAD 2016, Gefeller et al. Int J Environ Res Public Health 2016).

- l. 73: If the protocol has been developed using PRISMA-P, then this should be mentioned here. Additionally, adherence to PRISMA in reporting the systematic review should also be acknowledge here.

- l. 78: Restricting the search to three databases (PubMed, Cochran, Scopus) only poses some risk of missing relevant studies. You should give an explanation why such a restricted search strategy has been adopted and discuss it as a limitation.

- l. 112ff: You should only describe what you have actually done and not what you would have done. Only for one outcome you could perform a "mini" meta-analysis summarizing two studies and reporting a combined risk ratio. I did not find any (standardized) mean differences or odds ratios among the results that have been announced in your statistics section.

- l. 123: typo I^2 statistics (not statistical)

- l. 159-161: The decision to omit data from Darlow et al.'s study because "their declared p-values did not match the declared effects" implies a serious attack towards the integrity of Darlow et al.'s study publication. Did you contact the authors for clarification? You have to give more details on this issue. The reader must have the opportunity to understand your decision better.

- l. 170: shorter follow-up (instead of lower)

- l. 171: smaller population (instead of lower)

- l. 184: Your statement that "each study had a low risk of bias" contradicts what you have said before.

- l. 219: To my opinion "clinical practice" is the wrong term here, you should delete "clinical".

- l. 226/7: Changes of metabolism is a minor thread, reduction of vitamin D production is more relevant.

- l. 227: I guess you mean possible instead of feasible.

- l. 235: "utilization of mobile phones is lower" instead of "this may be a little lower".

General: You should add a section in the discussion commenting on the limitations of your systematic review (only RCTs, search restricted to only three databases, deletion of one study for data extraction, no meta-analysis for most outcomes etc.). You should also extend your discussion by reflecting on the future. Do you really think that further high-quality on this issue are needed (as you stated in your conlusion)? Use of text messaging via SMS is descending in all developed countries. There are better ways to reach individuals on the population level in order to promote sun protection (e.g. via messengers like Whats App that allow including pictures, videos, audio information or using smartphone apps).

6. PLOS authors have the option to publish the peer review history of their article (what does this mean?). If published, this will include your full peer review and any attached files.

Reviewer #1: Yes: Sherif M Badawy

Reviewer #2: No

---

## [Author Response · Author response to Decision Letter 0]

15 Jan 2020

Dear editor,

We appreciate very much the comments raised by you and the reviewers. Next, we will answer to each one:

Additional Editor Comments

E1C1: Abstract – why could results not be meta-analyzed?

We added an explanation regarding this topic in the Abstract as following: “however, they could not be meta-analyzed because outcomes were measured differently across studies”.

E1C2: Need to update PROSPERO page

We requested the update of the stage of this review in the PROSPERO page (https://www.crd.york.ac.uk/prospero/display_record.php?RecordID=91661). We requested a Yes in the Completed stage of the Preliminary searches, Piloting of the study selection process, Formal screening of search results against eligibility criteria, Data extraction, Risk of bias (quality) assessment, and Data analysis. 

E1C3: Please remove HBM acronym

HBM acronym is not present in the abstract of the study.

E1C4: Methods

Statistical analysis – this section needs correcting – lines 114-115 – “We present our results using mean differences (MD), risk ratios (RR), odds ratios (OR), and, when we meta-analyzed results from different studies for the same outcome that were measured using different scales, standardized mean differences (SMD).”

Meta-analysis was not SMD, but combined RR, MD and RR are not presented in the table only percentages for intervention and control groups or odds ratios – please be consistent and calculate OR for each outcome for each study to allow the reader to make comparisons. Be clear whether OR were adjusted or unadjusted.

Lines 117-118: There was only one outcome which was meta-analyzed. Were these methods on use of repeated measures used? It looks like it wasn’t as the outcome in Table 3 is “Sunburn anytime during follow-up (follow-up: from 5 to 12 months)”. Please ensure that the methods reflect exactly what you did in the analysis and how it was reported. 

As the editor states, in the methods section we detailed all the statistical analysis that we intended to perform before collecting the data (including what could happen with ORs, RRs, and SMDs). However, at the end we only performed one meta-analysis using RRs. In order to avoid confusion, we now only mention the statistical analysis that we are performing. This is stated in the Statistical analysis section of the Methods, as following:

“For each outcome of each study, we calculated and reported the intervention effects as mean differences (MDs) or risk ratios (RRs) along with their 95% confidence intervals (95% CIs).

When two or more studies presented the same outcome in a similar fashion, we performed a meta-analysis using random-effects models (Mantel-Haenszel method) due to heterogeneity across studies interventions (22). Meta-analyses were performed using Review Manager Software Version 5.3.

We assessed heterogeneity using the I2 statistics, and we considered that heterogeneity might not be important when I2 < 40% (23). Publication bias was not assessed since the number of studies pooled for each meta-analysis was lesser than ten (23).”.

Also, as the reviewer recommends, we added the RRs and MDs for each study to the Table 2 (Outcomes effects) in order to facilitate the reading of the results. We declared that these RRs and MDs were unadjusted at the bottom of the Table 2 (sentence: The risk ratios (RRs) and mean differences (MDs) were unadjusted).

E1C5: Was Darlow’s study actually excluded? If results were not considered to be reliable and were not used then it should be excluded and should not appear in Tables 2 & 3, the flow chart or anywhere else in the text or abstract. If you do choose to include it, then please be clear about the reasons for this.

The Darlow’s study presents results for the comparison of interest. However, the paper does not bring enough information regarding their results (they only present ANOVA F-test results for several groups, but not for the comparison of interest in our systematic review). We have mailed the authors asking for the results without any answer.

Since we are performing a comprehensive systematic review, we think that all studies that answer our question should be named, regardless of their omissions or lack of clarity. So, we are still showing its characteristics in table 1.

However, for table 2 (results of the studies), we now only mention the following: “The paper does not bring enough information in order to present its results”.

E1C6: There is a lot of information in Table 2 – it would be helpful if the main points were summarised in the text. 

p-values should be included in Table 2

We agree that data of statistical inference of each study is helpful for the readers. Thus, we have calculated the effect measures (MDs or RRs) and their 95% confidence intervals whenever possible (Table 2). We think that those measures are more useful than p-values for inference interpretation, and since this table is overloaded, we decided not to include p-values.

E1C7: Overall did text messages improve outcomes or not.

Consider a table with positive outcomes in green, negative in red and non-significant left white

Or a forest plot showing direction of effect and significance for each outcome without any pooling of data to give a visual picture of whether the intervention may be beneficial

In order to clarify the results of the included studies, we now present in Table 2 the RRs and MDs, and write in bold the statistically significant results.

E1C8: This needs more structure. Please provide quantitative results in the summary paragraph to report effect size and p-value. Please also clearly state the other outcomes reviewed and clarify that there was no compelling evidence to suggest that text messaging resulted in any improvements.

As suggested by the editor, we added the following lines showing the quantitative results in the Summary of the results section of the discussion:

“We only could meta-analyze one outcome (having a sunburn anytime during the follow-up), in which we did not find a statistically significant effect. Among the three studies that assessed sunscreen use, only one had a statistically significant effect (the one with the lowest population). Among the two studies that assessed a composite score of sun protection habits, one did not find an effect while the others find a small statistically significant difference. The certainty of the evidence was very low for these outcomes. Altogether, these studies do not show compelling evidence of any beneficial effect of text message reminders.”

E1C9: In the “Previous systematic reviews” section, please compare the outcomes assessed and findings of these reviews with those in the current study. What does this review offer over and above the previous reviews.

Thanks for this comment. We added the following lines to give more information regarding the previous reviews and delineate some comparisons, in the “Previous systematic reviews” section of the discussion: 

Although we did not find previous systematic reviews that have assessed the effects of SMS for sun protection habits, we found two previous systematic reviews that have assessed similar questions.

The first systematic review (11) assessed the effectiveness of SMS text reminders and similar electronic technology interventions to promote skin cancer prevention. This review describes the results of its included studies without performing any meta-analysis, and concluded that there was a lack of effect of their assessed interventions in skin cancer prevention outcomes. This conclusion is similar to ours, although our systematic review only included studies that have assessed SMS, and our outcome definition was broader.

The second systematic review (12) explored the effectiveness of SMS text reminders and mobile applications to improve adherence to preventive behaviors (sun protection, mental attention, the continuation of contraceptive pills, the use of condoms, among others) in adolescents. It included experimental and pre-post studies and did not formulate a clear conclusion regarding SMS for sun protection habits. 

E1C10: A limitations section is needed and needs to include the low quality of evidence, small numbers of studies and heterogeneity in outcome reporting. 

Thanks for this comment. We added the following paragraphs about Limitations:

“A possible limitation of this review is that search was performed in only three databases, which could cause that not all published studies were found. However, we manually searched potential studies for inclusion in the references of the included studies and also searched for studies that cited our included studies in Google Scholar; which could ensure that all relevant studies are included, even those from grey literature. Moreover, it has been evidenced that adding more databases to PubMed just increases the number of trials in 2.4% (49), and in most cases, it does not change the conclusion of the review (50). 

Additionally, the body of evidence presents important limitations: 1) the studies are heterogeneous in several aspects, such as the follow-up period (varies between one month and 12 months), which difficult the comparability of their results. In fact, the performed meta-analysis pooled the result of a 5-months follow-up with the result of a 12-months follow-up. 2) Most of the outcomes were not measured in the same way. 3) Most studies measured outcomes as self-report, introducing a recall bias. 4) The studies had a high risk of bias or some concerns in several domains. 5) Overall, the certainty of the evidence was low in the main outcomes, mainly due to the risk of bias, inconsistency, and small sample size.

Reviewer 1:

R1C1: This is an interesting systematic review that examined the evidence for text messaging as an intervention to promote adherence to sun protection strategies. Below are concerns/comments for the authors to consider.

Thank you very much for your kind words.

R1C2: The authors should clarify in the abstract and the methods section if they followed reporting guidelines, such as PRISMA, or other guidelines. Include appropriate citations as well.

&

R2C4: l. 73: If the protocol has been developed using PRISMA-P, then this should be mentioned here. Additionally, adherence to PRISMA in reporting the systematic review should also be acknowledge here.

Thank you for the comments. We added the following sentence in the Abstract: We performed a systematic search in PubMed, Central Cochrane Library, and Scopus, following the PRISMA recommendations to perform systematic reviews.

We also added the following sentence in the Methods section: We performed a systematic review and meta-analysis following the Preferred Reporting Items for Systematic Reviews and Meta-analyses (PRISMA) recommendations (19).

R1C3: Introduction section, to support the rationale for the review, the authors should include additional recent promising evidence that support feasibility, acceptability and efficacy of digital interventions for behavior change (References - PMID: 30026178; PMID: 25803705; PMID: 28506955; PMID: 26831740; PMID: 28428157; PMID: 26701961; PMID: 29273573).

Thank you for the comments. We added the following lines in the second paragraph of the Introduction citing the suggested literature: 

“In this way, there is a growing interest in technology-based interventions, such as the use of mobile applications (8), electronic mails (9) and short message service text message reminders (SMS text reminders) (10, 11). There are some systematic reviews that have synthetized the evidence of the effects of SMS text reminders and mobile applications in medication adherence and management in adolescents (12-14) and adults (15), supporting their feasibility and acceptability. Nevertheless, all of those reviews recommended future studies with a more fitting design.”

R1C4: Results, the authors should include much more details for the included studies, such as age, gender, study settings, follow up, outcomes, intervention details, and other RCT related factors (blinding, randomization, etc.). Similar details should be also added to Table 1.

We added the suggested information in:

- Characteristics section (Results): 

o “Regarding the population characteristics, three studies (27-29) were performed in community dwellers; the female proportion ranged from 8.4% to 100%, and the mean age ranged from 31.6 to 44.2 years. 

o Regarding the intervention, it consisted on delivering SMS text reminders on sun protection habits, such as sunscreen use and skin self-examination, with heterogeneous frequency (range: three to seven days) and duration (range: one to 12 months). Of the five RCTs, two of them tailored the messages according to baseline characteristics of participants, using the health belief model (HBM) constructs (24, 25).”

- Risk of bias section (Results): 

o “Regarding the randomization process, three studies had some concerns, while three studies had high risk of bias in the measurement of the outcome. Only one study had low risk of bias in most domains (29). Four studies had high risk of bias in the overall judgment (Figure 2).”

- Outcome effects (Results): 

o “Several objective outcomes were reported by the included studies. Some of them were the number of sunburns, sunscreen use, sun protection habits (protecting clothes, wearing sunglasses, wearing a hat, etc.), skin self-examination, attempted suntan and adherence rate of sunscreen application.”

We are reporting the results of the included studies in Table 2. And we also added the suggested variables in the Table 1: study settings, follow-up period, and intervention details.

R1C5: All the figures are fuzzy and unclear. Replace with more clear ones.

We apologize for it. We are submitting all the three figures in JPG format.

R1C6: Discussion, the authors should expand and elaborate more on how their findings support or contrast available literature and provide suggestions for future research directions that would address existing knowledge gaps.

The comparison with previous literature has been improved, as detailed in the answer to E1C9.

To provide suggestion for future research directions, we added the following in the “implications for practice” section of the discussion: 

“However, researchers must reflect on the need for performing more studies using SMS, since it could be descending in favor of other messengers like WhasApp. On the other hand, nowadays the ubiquitous use of smartphones may allow to use other tools such as ad-hoc mobile applications (which allows face-to-face interactions, a more friendly interaction, and using videos/images) or messengers like WhatsApp (which allows including pictures, videos, audio information) (47, 48). Given these alternatives, maybe studies using SMS should be limited to those contexts where smartphones use is still low.”

R1C7: Discussion, it is critical to discuss the value of including direct patients' input in the development of mhealth interventions and other key considerations for end users should be sought early on in the process of app or digital behavioral intervention design to ensure long and short term engagement (PMID: 29273573; PMID: 26844685; PMID: 27966189; PMID: 28241759).

Thank you for suggesting literature to cite. We added the following lines to discuss the participants’ input in the Characteristics of SMS text reminders section: 

“Personalization of the SMS content is an important element to ensure the engagement with the intervention (30), since it may influence the attitudes, motivation, and attention of the participants (31). Among our included studies, two studies (Darlow and Gold) (25, 28) developed the intervention messages using focus group feedback, while the Youl study (26) collected a pilot survey to estimate the participants’ preferences of the SMS text messages content. 

For tailoring, two studies (Darlow and Duffy) (24, 25) used the HBM profile of their participants to create the messages. The HBM profile consists of four constructs: perceived susceptibility to ill-health, perceived severity of ill-health, perceived benefits of behavior change, and perceived barriers to taking action (32). Since these psychological aspects can influence the participant’s perception of the message, tailored messages using the HBM profile are thought to have a greater impact.”

Moreover, we added a column to describe the Participants’ input in the Supplementary material 3. 

R1C8: Discussion, the authors should also acknowledge the lack of economic data to support the use of mhealth behavioral interventions to date (PMID: 27780795; PMID: 28152012).

We agree with the reviewer, and accordingly we have added the following lines in the “Implications for practice” section: 

“Regarding resource use, systematic reviews of economic evaluations of text messaging interventions found no comprehensive evidence (42, 43). Particularly, cost studies of SMS reminders for improving sun protection habits are needed (33), such as those performed in SMS interventions for other topics such as diabetes mellitus prevention (44), improvement of antiretroviral therapy adherence (45), and smoking cessation (46).”

Reviewer 2:

R2C1: The manuscript describes conduct and results of a systematic review of RCTs adressing the effect of SMS text reminders in promoting sun protection measures. Overall, the manuscript represents sound work.

Thank you very much.

R2C2: l. 50-52: Avoidance of exposure during peak UV hours around solar noon is mentioned by all guidelines promoting sun protection and should be included in the list of sun protection habits.

Thank you for the recommendation. We included the phrase “avoidance of exposure during peak UV hours around solar noon” in the first paragraph of Introduction.

R2C3: l. 53: There are a lot more studies evaluating compliance with recommendations for sun protection. You should add recent papers (e.g. Vogel et al. Cancer Epidemiol Biomarkers Prev 2017, Barkin et al. JAAD 2016, Gefeller et al. Int J Environ Res Public Health 2016).

Thank you for suggesting references. We added the following sentence in the first paragraph of Introduction: 

“Accordingly, educational interventions to enhance compliance with sun protection habits have been proposed, and considerable effects have been observed in certain groups, such as melanoma survivors, parents of young children, and medical professionals (4-6).”

R2C5: l. 78: Restricting the search to three databases (PubMed, Cochran, Scopus) only poses some risk of missing relevant studies. You should give an explanation why such a restricted search strategy has been adopted and discuss it as a limitation.

We agree that this could be a limitation of our study. So, we have added the following paragraph to the limitations section of the discussion: 

“A possible limitation of this review is that search was performed in only three databases, which could cause that not all published studies were found. However, we manually searched potential studies for inclusion in the references of the included studies and also searched for studies that cited our included studies in Google Scholar; which could ensure that all relevant studies are included, even those from grey literature. Moreover, it has been evidenced that adding more databases to PubMed just increases the number of trials in 2.4% (49), and in most cases, it does not change the conclusion of the review (50).”

R2C6: l. 112ff: You should only describe what you have actually done and not what you would have done. Only for one outcome you could perform a "mini" meta-analysis summarizing two studies and reporting a combined risk ratio. I did not find any (standardized) mean differences or odds ratios among the results that have been announced in your statistics section.

We agree with this commentary, and we have simplified our statistical analysis section accordingly, as detailed in the answer to E1C4.

R2C7: l. 123: typo I^2 statistics (not statistical)

Thank you for the correction. We are now using the term “I2 statistics” in the methods section

R2C8: l. 159-161: The decision to omit data from Darlow et al.'s study because "their declared p-values did not match the declared effects" implies a serious attack towards the integrity of Darlow et al.'s study publication. Did you contact the authors for clarification? You have to give more details on this issue. The reader must have the opportunity to understand your decision better.

The Darlow’s study presents results for the comparison of interest. However, the paper does not bring enough information regarding their results (they only present ANOVA F-test results for several groups, but not for the comparison of interest in our systematic review). We have mailed the authors asking for the results without any answer.

Since we are performing a comprehensive systematic review, we think that all studies that answer our question should be named, regardless of their omissions or lack of clarity. So, we are still showing its characteristics in table 1.

However, for table 2 (results of the studies), we now only mention the following: “The paper does not bring enough information in order to present its results”

R2C9: l. 170: shorter follow-up (instead of lower)

Thank you for the correction. We are now using the phrase “shorter follow-up”.

R2C10: l. 171: smaller population (instead of lower)

Thank you for the correction. We are now using the term “smaller population”. 

R2C11: l. 184: Your statement that "each study had a low risk of bias" contradicts what you have said before.

Thank you for the correction. We replaced that sentence with the following one in Summary of the results section (Discussion): In the overall assessment, four studies had a high risk of bias and one study had some concerns.

R2C12: l. 219: To my opinion "clinical practice" is the wrong term here, you should delete "clinical".

Thank you for the correction. We are now using the phrase “Implications for practice”.

R2C13: l. 226/7: Changes of metabolism is a minor thread, reduction of vitamin D production is more relevant.

Thank you for this important observation. We added the following lines in the second paragraph of Implications for practice section: 

“For example, a study in 3,194 Danish found that seeking shade and wearing protective clothing was significantly associated with lower vitamin D levels in adults (37), and an analysis of a representative survey in USA (US National Health and Nutrition Examination Survey 2003–2006) found that staying in the shade and wearing long sleeves were significantly associated with lower 25(OH)D levels, especially in individuals who reported frequent use of shade on sunny days (38).”

R2C14: l. 227: I guess you mean possible instead of feasible.

Thank you for the correction. We are now using the term “possible”.

R2C15: l. 235: "utilization of mobile phones is lower" instead of "this may be a little lower".

Thank you for the correction. We are now using the phrase “utilization of mobile phones is lower”.

R2C16: General: You should add a section in the discussion commenting on the limitations of your systematic review (only RCTs, search restricted to only three databases, deletion of one study for data extraction, no meta-analysis for most outcomes etc.).

We have added a paragraph on the limitations section of the discussion regarding this topic:

“Additionally, the body of evidence presents important limitations: 1) the studies are heterogeneous in several aspects, such as the follow-up period (varies between one month and 12 months), which difficult the comparability of their results. In fact, the performed meta-analysis pooled the result of a 5-months follow-up with the result of a 12-months follow-up. 2) Most of the outcomes were not measured in the same way. 3) Most studies measured outcomes as self-report, introducing a recall bias. 4) The studies had a high risk of bias or some concerns in several domains. 5) Overall, the certainty of the evidence was low in the main outcomes, mainly due to the risk of bias, inconsistency, and small sample size.”

R2C17: You should also extend your discussion by reflecting on the future. Do you really think that further high-quality on this issue are needed (as you stated in your conlusion)? Use of text messaging via SMS is descending in all developed countries. There are better ways to reach individuals on the population level in order to promote sun protection (e.g. via messengers like Whats App that allow including pictures, videos, audio information or using smartphone apps).

Thank you for the observation, this is definitely an important issue to be discussed. Accordingly, we have added the following lines in the Implications for practice section of the discussion: 

“However, researchers must reflect on the need for performing more studies using SMS, since it could be descending in favor of other messengers like WhasApp. On the other hand, nowadays the ubiquitous use of smartphones may allow to use other tools such as ad-hoc mobile applications (which allows face-to-face interactions, a more friendly interaction, and using videos/images) or messengers like WhatsApp (which allows including pictures, videos, audio information) (47, 48). Given these alternatives, maybe studies using SMS should be limited to those contexts where smartphones use is still low.”

---

## [Decision Letter · Decision Letter 1]

26 Feb 2020

PONE-D-19-24279R1

Text message reminders for improving sun protection habits: a systematic review

PLOS ONE

Dear Dr Taype-Rondan,

Thank you for submitting your manuscript to PLOS ONE. After careful consideration, we feel that it has merit but does not fully meet PLOS ONE’s publication criteria as it currently stands. Therefore, we invite you to submit a revised version of the manuscript that addresses the points raised during the review process.

Please address the final points by the Academic Editor and proofread for readability.

We would appreciate receiving your revised manuscript by 17 April 2020. To enhance the reproducibility of your results, we recommend that if applicable you deposit your laboratory protocols in protocols.io, where a protocol can be assigned its own identifier (DOI) such that it can be cited independently in the future. For instructions see: http://journals.plos.org/plosone/s/submission-guidelines#loc-laboratory-protocols

We look forward to receiving your revised manuscript.

Kind regards,

Jennifer A Hirst, DPhil

Academic Editor

PLOS ONE

Additional Editor Comments (if provided):

There are still some minor points which need to be revised before this paper is suitable for publication:

Please update PROSPERO – it still says only preliminary searches have been conducted.

Please remove HBM acronym throughout the article. This acronym is unnecessary

Statistical analysis section:

Change ..” lesser than ten” to “less than ten”

Results:

Please include reference numbers for each of the studies in tables 1-3.

Change title of table 2 to: Characteristics of included studies

Limitations:

Incorrect grammar: “A possible limitation of this review is that search was performed in only three databases, which could cause that not all published studies were found.”

Typo: “...such es the follow-up period”

We recommend asking a native English speaker to proofread the article.

Reviewers' comments:

Reviewer's Responses to Questions

**Comments to the Author**

1. If the authors have adequately addressed your comments raised in a previous round of review and you feel that this manuscript is now acceptable for publication, you may indicate that here to bypass the “Comments to the Author” section, enter your conflict of interest statement in the “Confidential to Editor” section, and submit your "Accept" recommendation.

Reviewer #1: All comments have been addressed

Reviewer #2: All comments have been addressed

2. Is the manuscript technically sound, and do the data support the conclusions?

Reviewer #1: Yes

Reviewer #2: Yes

3. Has the statistical analysis been performed appropriately and rigorously? 

Reviewer #1: Yes

Reviewer #2: Yes

4. Have the authors made all data underlying the findings in their manuscript fully available?

Reviewer #1: Yes

Reviewer #2: Yes

5. Is the manuscript presented in an intelligible fashion and written in standard English?

Reviewer #1: Yes

Reviewer #2: No

6. Review Comments to the Author

Reviewer #1: The authors addressed all my earlier concerns. No additional comments. This review contributes to existing literature on tech-based behavioral interventions.

Reviewer #2: Final proofreading of the text should be done by a native speaker who can correct language errors.

7. PLOS authors have the option to publish the peer review history of their article (what does this mean?). If published, this will include your full peer review and any attached files.

Reviewer #1: Yes: Sherif M Badawy

Reviewer #2: No

---

## [Editor Report · Decision Letter 2]

12 Mar 2020

PONE-D-19-24279R2

Text message reminders for improving sun protection habits: a systematic review

PLOS ONE

Dear Dr Taype-Rondan,

Thank you for submitting your manuscript to PLOS ONE. After careful consideration, we feel that it has merit but does not fully meet PLOS ONE’s publication criteria as it currently stands. Therefore, we invite you to submit a revised version of the manuscript that addresses the points raised during the review process. Specifically to correct the English. We recommend that the paper is proof-read by a native English speaker.

We would appreciate receiving your revised manuscript by 1 May 2020. To enhance the reproducibility of your results, we recommend that if applicable you deposit your laboratory protocols in protocols.io, where a protocol can be assigned its own identifier (DOI) such that it can be cited independently in the future. For instructions see: http://journals.plos.org/plosone/s/submission-guidelines#loc-laboratory-protocols

We look forward to receiving your revised manuscript.

Kind regards,

Jennifer A Hirst, DPhil

Academic Editor

PLOS ONE

Additional Editor Comments (if provided):

The English is still poor. Please change: "Due to the fact that search was only performed in three databases, we might not found all published studies" to "Due to the fact that search was only performed in three databases, we might not have found all published studies."

---

## [Author Response · Author response to Decision Letter 2]

27 Mar 2020

Dear editor,

We appreciate very much the comments raised by you and the reviewers. We are answering to each one in the attached file.

---

## [Editor Report · Decision Letter 3]

1 May 2020

Text message reminders for improving sun protection habits: a systematic review

PONE-D-19-24279R3

Dear Dr. Taype-Rondan,

We are pleased to inform you that your manuscript has been judged scientifically suitable for publication and will be formally accepted for publication once it complies with all outstanding technical requirements.

With kind regards,

Jennifer A Hirst, DPhil

Academic Editor

PLOS ONE
---

## [Editor Report · Acceptance letter]

7 May 2020

PONE-D-19-24279R3 

Text message reminders for improving sun protection habits: a systematic review 

Dear Dr. Taype-Rondan:

I am pleased to inform you that your manuscript has been deemed suitable for publication in PLOS ONE. Congratulations! Your manuscript is now with our production department. 

With kind regards,

on behalf of

Dr. Jennifer A Hirst 

Academic Editor

PLOS ONE